# Semi-supervised Visible-Infrared Person Re-identification via Modality Unification and Confidence Guidance

## ABSTRACT

Semi-supervised visible-infrared person re-identification (SSVI-ReID) aims to match pedestrian images of the same identity from different modalities (visible and infrared) while only annotating visible images, which is highly related to multimedia and multi-modal processing. Existing works primarily focus on assigning accurate pseudo-labels to infrared images, but overlook the two key challenges: erroneous pseudo-labels and large modality discrepancy. To alleviate these issues, this paper proposes a novel Modality-Unified and Confidence-Guided (MUCG) semi-supervised learning framework. Specifically, we first propose a Dynamic Intermediate Modality Generation (DIMG) module, which transfers knowledge from labeled visible images to unlabeled infrared images, enhancing the pseudo-label quality and bridging the modality discrepancy. Meanwhile, we propose a Weighted Identification Loss (WIL) that can reduce the model's dependence on erroneous labels by using confidence weighting. Moreover, an effective Modality Consistency Loss (MCL) is proposed to narrow the distribution of visible and infrared features, further narrowing the modality discrepancy and enabling the learning of modality-unified features. Extensive experiments show that the proposed MUCG has significant advantages in improving the performance of the SSVI-ReID task, surpassing the current state-of-the-art methods by a significant margin. The code will be available.

## CCS CONCEPTS

• **Information systems → Information retrieval**.

## KEYWORDS

Modality unification, Confidence guidance, Semi-supervised learning, VI-ReID

## 1 INTRODUCTION

Traditional person re-identification (ReID) [15, 17, 63] refers to matching pedestrian images with the same identity captured from non-overlapping visible cameras. Existing cameras include two modalities: visible and infrared. In low-light scenarios, cameras will automatically switch from visible modality to infrared modality. However, in nighttime or low-light environments, the pedestrian images captured by visible cameras cannot obtain effective appearance information, which hinders the applicability of ReID

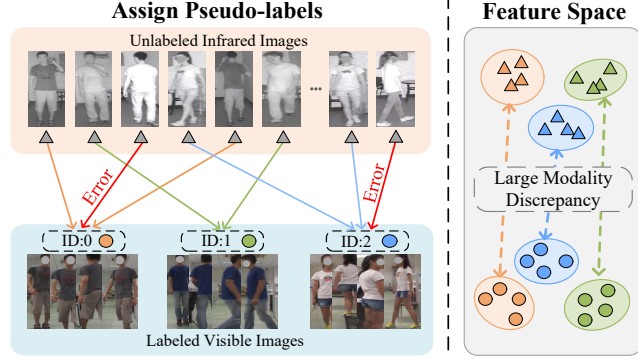

**Figure 1: Illustration of two critical factors affecting the performance of SSVI-ReID: erroneous pseudo-labels and large modality discrepancy.**

in practice. Therefore, [40] propose a challenging cross-modality visible-infrared ReID (VI-ReID) task.

The cross-modality VI-ReID task [10, 23, 50] solves the problem of person ReID under poor lighting conditions, aiming to match nighttime infrared person images captured by infrared cameras with visible person images. At present, significant progress has been made in VI-ReID. Some widely used VI-ReID techniques [35, 40] strive to identify distinct embedding spaces that minimize the gap between different modalities at the embedding level. Nevertheless, the significant modality gap poses a challenge for these methods in locating appropriate embedding spaces. Alternatively, there are image-level approaches [9, 56] that aim to transform images from one modality to another, effectively bridging the modality gap between visible and infrared images. Despite their success in reducing the modality gaps, the generated cross-modality images are usually accompanied by some noises. An important factor for the above methods to achieve good results is their well-annotated cross-modality training sets. However, annotating cross-modality ReID data is extremely time-consuming and requires extremely high costs. Additionally, the lack of color information in infrared images makes it more difficult to annotate cross-modality images manually. These problems motivate us to train a cross-modality ReID model using labeled visible data and unlabeled infrared data.

Therefore, the investigation of the semi-supervised VI-ReID (SSVI-ReID) task holds significant importance. It aims to learn modality-invariant knowledge from labeled visible data and unlabeled infrared data, thereby achieving cross-modality pedestrian image retrieval. However, existing single-modality UDA-ReID methods (using labeled visible images as the source domain and unlabeled infrared images as the target domain) suffer from cross-modality discrepancy, making it difficult to directly learn modality invariant features. Besides, the current semi-supervised VI-ReID

methods [33, 37] primarily focus on how to correctly assign pseudo-labels. OTLA [37] focuses on the assignment of infrared pseudo-labels. DIPS [33] generates pseudo labels dependently on multi-model collaboration, which might lead to reduced efficiency. They often neglect the negative impact of noisy pseudo-labels and modality discrepancy. Therefore, as shown in Figure 1, how to eliminate the negative impact of noisy pseudo-labels and transfer the learned knowledge from visible modality to infrared modality under semi-supervised settings is the key to the SSVI-ReID task.

In this paper, we propose a new Modality-Unified and Confidence-Guided (MUCG) semi-supervised method for VI-ReID without the labels of infrared images. To address the issue of noisy labels and the modality discrepancy between the labeled visible and unlabeled infrared images, we propose the following three modules. Firstly, we propose a Dynamic Intermediate Modality Generation (DIMG) module that generates intermediate modality features by mixing the features of visible and infrared modalities. Using intermediate modality features to improve the discriminative ability of the model for unlabeled infrared images. Secondly, to reduce the negative impact of noisy pseudo-labels, we propose a Weighted Identification Loss (WIL) to calculate the confidence of pseudo-labels. By assigning different weights to different pseudo-labels, the WIL can ensure that the model pays more attention to reliable labels during the training process, while reducing dependence on unreliable labels. Finally, to address the issue of cross-modalities discrepancy, we propose an effective Modality Consistency Loss (MCL) to minimize the distances between visible and infrared modalities. The three modules, DIMG, WIL, and MCL focus on enhancing the model's adaptability to modality differences, reducing the impact of noisy labels, and enhancing feature alignment, respectively, thus solving the issues of noisy labels and modality discrepancies. The proposed method significantly improves the overall performance of the model in the SSVI-ReID task. Specifically, the MUCG method achieves a Rank-1 accuracy of 68.8% on the SYSU-MM01 dataset, 86.9% on the RegDB dataset, and 51.9% on the LLCM dataset, surpassing the current state-of-the-art semi-supervised methods.

The main contributions can be summarized as follows:

(1) We propose a novel modality-unified and confidence-guided semi-supervised VI-ReID framework that exclusively relies on the annotation of visible images, offering a cost-effective solution.

(2) We design a dynamic intermediate modality generation module, which can effectively enhance the model's discriminative ability of unlabeled infrared images.

(3) We propose a weighted identification loss and a modality consistency loss, alleviating the negative impact of noisy pseudo-labels and narrowing the modality gap between visible and infrared.

(4) The proposed method outperforms other state-of-the-art methods for the semi-supervised VI-ReID task on three challenging datasets, as demonstrated by extensive experiments.

## 2 RELATED WORK

### 2.1 Supervised Visible-Infrared Person ReID

Supervised visible-infrared person ReID (SVI-ReID) aims to match infrared images with visible images of pedestrians under non-overlapping cameras. Recently, some works [21, 42, 57] try to mine

modality-invariant information by using complex network structures or generation methods to alleviate modality discrepancy. [40] starts the first attempt by proposing a zero-padding one-stream network toward automatically evolving modality-specific nodes. [11] utilize the modality-sharing layer to develop shared knowledge and improve the modality invariance of deep representation. Additionally, a channel enhancement (CA) method is introduced in [47] to uniformly generate color-independent images by randomly swapping color channels.

Although the supervised VI-ReID methods mentioned above have achieved good results, they require a large amount of cross-modality identity annotations, which hinders the rapid deployment of new scenes. Manual annotation requires a high cost, especially for infrared images. In this work, we investigate the semi-supervised visible-infrared person ReID task, which does not require infrared identity annotation and is of great significance for deploying VI-ReID in the real world.

### 2.2 Unsupervised Domain Adaptation Person ReID

The goal of unsupervised domain adaptation (UDA) is to enhance learning of the unlabeled target domain through labeled source domains. It can be roughly divided into three categories, *i.e.* fine-tuning [2, 5], GAN transferring [8, 18, 39], and joint training [6, 13, 60]. Fine-tuning methods first train the model using labeled source data and then fine-tune the pre-trained model on the target data using pseudo-labels [58]. GAN transfer methods disentangle features into id-related and id-unrelated features [64] or use GAN to transfer the style of images [8]. Joint training methods combine the source data and target data and use the ImageNet network to train from scratch [20]. However, these methods ignore the bridging between two domains, that is, using the similarity between the two domains to learn domain invariant information.

The task of this paper is similar to unsupervised domain adaptive ReID [37, 43]. Labeled visible images are the source domain and unlabeled infrared images are the target domain. UDA-VI-ReID aims to transfer learned knowledge from labeled visible images to unlabeled visible infrared images and match images of the same person captured by both visible and infrared cameras. In addition, the unsupervised domain adaptation ReID task is a homogeneous retrieval task, while the semi-supervised VI-ReID task is a heterogeneous retrieval task. The domain difference between visible and infrared images is greater than that in the UDA ReID task, making it a significant challenge.

### 2.3 Pseudo-labels in Semi-supervised Learning

The pseudo-labeling method is a supervised paradigm that learns from both unlabeled and labeled data simultaneously which uses the class with the highest prediction probability as the pseudo-label. According to the assumption of semi-supervised learning [1, 24, 25], the decision boundary should pass through areas with sparse data to avoid dividing dense sample data points on both sides of the decision boundary. This means that the model needs to make low entropy predictions on unlabeled data, *i.e.* minimizing entropy. Pseudo-labels can effectively reduce class overlap, leading to clearer class boundaries and more compact learned classes.

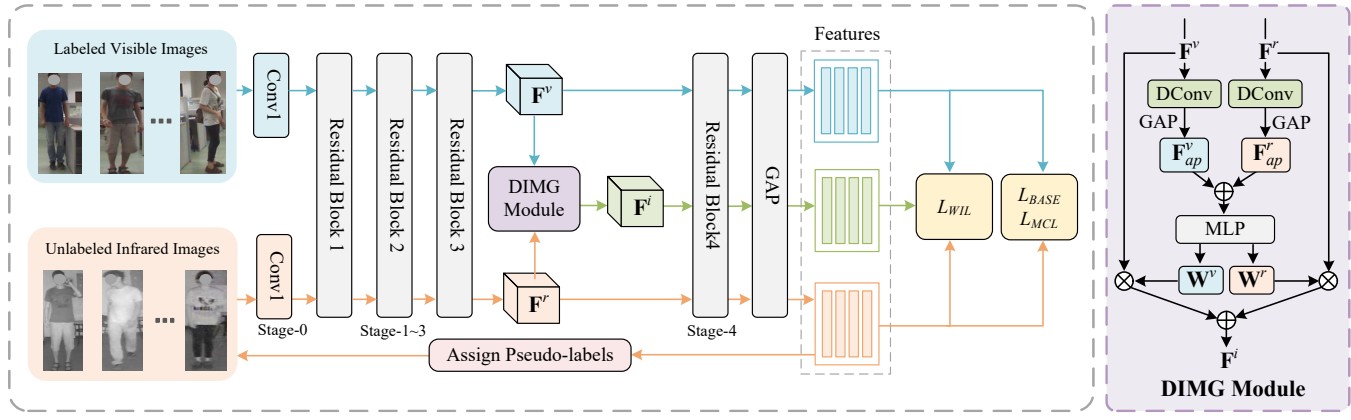

Figure 2: Framework of the proposed MUCG. MUCG adopts independent blocks in stage-0 to extract low-level visible and infrared features and the remaining stages are utilized as modality-shared ResBlocks. The DIMG module is used to generate intermediate modality features, serving as an intermediate bridge between visible and infrared features, and improving the model's recognition ability for infrared images. The proposed method utilizes the original visible and infrared features as well as intermediate features during training, and incorporates them into our objective function consisting of $L_{BASE}$, $L_{WIL}$, and $L_{MCL}$. "DConv" means depth-wise convolution. "GAP" means global average pooling. "MLP" refers to multilayer perceptron.

UPS [32] proposal of high confidence pseudo-labels may not necessarily be correct, while low confidence pseudo-labels are basically incorrect. Based on the above content, when selecting a subset of pseudo-label predictions, we choose high-confidence predictions as positive examples and low-confidence predictions as negative examples. Self-tuning method [38] proposes using a pseudo-label group comparison mechanism to mitigate the impact of noisy labels. FixMatch [34], ConMatch [22], and FlexMatch [52] all use thresholds to select high-confidence pseudo-labels for training.

In addition, [37] formulates the label assignment task as an optimal transportation (OT) problem, treating unlabeled samples as suppliers and pseudo-labels as demands. Through the optimal transportation plan, the supplier samples are transported to the demand side at the lowest cost. In this paper, we apply OT to the infrared data label allocation problem. This method can force infrared samples to be assigned to equally sized subsets, avoiding grouping samples together. Furthermore, the quality of pseudo-labels is closely related to the calibration error (i.e. the predictive ability) of the model. This paper proposes an effective WIL to reduce the impact of erroneous pseudo-labels on the model.

## 3 METHODOLOGY

In this section, we first introduce the model architecture of the proposed Modality-Unified and Confidence-Guided (MUCG) semi-supervised VI-ReID. Then, we elaborate on the design of the Dynamic Intermediate Modality Generation (DIMG) Module, Weighted Identification Loss (WIL), and Modality Consistency Loss (MCL) in detail. Finally, we adopt a multi-loss strategy to jointly optimize the proposed semi-supervised VI-ReID method.

### 3.1 Model Architecture

Figure 2 provides an overview of the proposed MUCG method. The inputs of MUCG are labeled visible images and unlabeled infrared

images, which are fed into the DIMG module to generate intermediate modality features. Under the semi-supervision setting, we can only access the labels $\mathbf{Y}^v$ of visible images. For unlabeled infrared images, we initially randomly generate pseudo-labels for them. Then, we introduce the optimal transport assignments [37, 58] to update pseudo-labels,

$$\mathbf{P}^* = \text{diag}(\alpha)\mathbf{P}^\gamma\text{diag}(\beta), \tag{1}$$

where $\text{diag}(\cdot)$ denotes the square diagonal matrix with the elements of vector on the main diagonal, $\mathbf{P}$ is the softmax output of the infrared image classifier, $\gamma$ is a parameter that controls the smoothness of the mapping, $\alpha$ and $\beta$ represent class prior uniform distribution vector and sample prior uniform distribution vector respectively. Through them, it is possible to force the assignment of infrared samples to equally sized subsets. The infrared pseudo-labels $\mathbf{Y}^r$ are as follows,

$$\mathbf{Y}^r = \text{argmax}(\mathbf{P}^*), \tag{2}$$

where $\text{argmax}(\cdot)$ is used to find the index of the maximum value in each row of $\mathbf{P}^*$, determine the most likely category of each sample, thereby generating an infrared pseudo-label $\mathbf{Y}^r$.

Inspired by the work of PCB [36] in extracting discriminative features, we horizontally divide the feature map $\mathbf{F}_g$ into three parts $\{\mathbf{F}_{p1}, \mathbf{F}_{p2}, \mathbf{F}_{p3}\}$, each of which is fed into the classifier to learn local knowledge. In addition, to reduce the modality discrepancy and eliminate the negative impact of noisy pseudo-labels, we propose a novel Weighted Identification Loss (WIL) and a Modality Consistency Loss (MCL).

### 3.2 Dynamically Intermediate Modality Generation Module

Unlike unsupervised visible ReID problems, visible and infrared images have significant appearance discrepancies in the SSVI-ReID task. We draw inspiration from works [6, 62], which show that

adding an intermediate domain as the bridge can better transfer knowledge from the source domain to the target domain. Therefore, we introduce an intermediate modality as a bridge to transfer labeled visible modality knowledge to the unlabeled infrared modality, improving the model's ability to distinguish infrared images.

As shown in Figure 2, we generate intermediate modality features by mixing visible and infrared features. The DIMG module we proposed can be inserted after the hidden stage in the backbone network . This module takes the output features $(\mathbf{F}^v, \mathbf{F}^r)$ of visible and infrared images $(\mathbf{X}^v, \mathbf{X}^r)$ in stage-3 as input and generates two weight factors $(\mathbf{W}^v, \mathbf{W}^r)$. We can mix visible and infrared features with these two weighting factors to dynamically generate intermediate modality features.

In each mini-batch during the training stage, we combine samples into n sample pairs based on labels. For each sample pair $(\mathbf{X}^v, \mathbf{X}^r)$, both samples have the same label (pseudo-label). After obtaining their feature maps $\mathbf{F}^v, \mathbf{F}^r \in \mathbb{R}^{h \times w \times c}$, we use the large convolution kernel of depth-wise convolution to extract discriminative features from the visible and infrared modalities. Following [4], we set the kernel size to 63. Then, we apply average-pooling to both features, resulting in $1 \times 1 \times c$ dimensional features $(\mathbf{F}^v_{ap}, \mathbf{F}^r_{ap})$, and their output feature vectors are summed and inputted into $MLP$ consisting of two fully connected layers to generate two weighting factors:

$$[\mathbf{W}^v, \mathbf{W}^r] = \delta(MLP(\mathbf{F}^v_{ap} + \mathbf{F}^r_{ap})), \tag{3}$$

where $\delta(\cdot)$ is a softmax function, $\mathbf{W}^v$ and $\mathbf{W}^r$ are weighting factors for visible and infrared features, respectively. Weighting factors are used to dynamically fuse the features of two modalities. Therefore, the formula for generating intermediate modality features can be written as follows:

$$\mathbf{F}^i = \mathbf{W}^v \times \mathbf{F}^v + \mathbf{W}^r \times \mathbf{F}^r. \tag{4}$$

Then, the intermediate modality features and original features are fed together into the network.

Our proposed DIMG module can learn in an effective joint training scheme, rather than undergoing arduous training on GANs or reconstructed images. By utilizing appropriate intermediate modalities to connect the visible and infrared domains, visible knowledge can be better transferred to the infrared domain and improve the discriminative ability of the model in the infrared domain. However, relying solely on the DIMG module is not enough to fully address all the challenges in the SSVI-ReID task. Especially in small datasets, the problem of noisy labels during training has become a challenge that we must face. To address this challenge, we further propose weighted identification loss.

### 3.3 Weighted Identification Loss

Unlike other semi-supervised learning methods [22, 34, 52] that only select high-confidence samples during the sample selection stage, we use all samples for training due to the small size of the VI-ReID datasets. However, the inevitable inclusion of noisy labels in pseudo-labeled samples can significantly reduce model performance. To alleviate this issue, we propose a Weighted Identification Loss (WIL) that utilizes confidence weighting to mitigate the impact of incorrect labels. Drawing inspiration from work [44], we utilize the memory effect of deep neural networks (DNN) to calculate the

correct labeling confidence for each sample by simulating the loss distribution. The loss distribution of each sample in all training data is fitted by a two-component Gaussian mixture model, as shown below:

$$p(L^{id}|\theta) = \sum_{k=1}^{K} \eta_k \varphi(L^{id}|k), \tag{5}$$

where $\eta_k$ and $\varphi(L^{id}|k)$ are the mixture coefficient and probability density of the $k$-th component, respectively. $L^{id}$ is the identification (cross-entropy) loss. Based on the memory effect of DNN, we can calculate the correct annotation confidence $w^k$ for each sample $k$:

$$w_k = p(m|L^{id}_k), \tag{6}$$

where $m$ is the posterior probability over the small mean value component. Therefore, the proposed WIL can be expressed as follows:

$$L_{WIL-} = -\frac{1}{K} \sum_{k=1}^{K} w_k log(p(y_k|x_k)), \tag{7}$$

where $x_k$ is the input image feature, $y_k$ is the corresponding label, and $p(y_k|x_k)$ is the prediction probability that $x_k$ is recognized as class $y_k$. However, as pointed out in [32], low-confidence pseudo-labels are largely incorrect, so we set a certain threshold. When the confidence is below this threshold, the sample is treated as a negative sample for learning. So, the proposed WIL is as follows:

$$L_{WIL} = -\frac{1}{K} \sum_{k=1}^{K} (w^p_k log(p(y_k|x_k)) + w^n_k log(1-p(y_k|x_k))), \tag{8}$$

where

$$\begin{cases} w^p_k = w_k, & w^n_k = 0, & w_k > \tau \\ w^p_k = 0, & w^n_k = 1, & \text{otherwise} \end{cases}, \tag{9}$$

$\tau$ is a threshold for positive and negative labels, and we set it to 0.1. $w^p_k$ is the positive learning weight, and $w^n_k$ is the negative learning weight. For visible images, since their labels are known and correct, we set $w_k$ to 1. For infrared images, the proposed WIL can enable all pseudo-label samples to play a role in the training process while more accurately evaluating the confidence of pseudo-labels and weighting the loss function accordingly, reducing the negative impact of noisy labels on model training.

### 3.4 Modality Consistency Loss

Despite WIL's ability to optimize the model's handling of noisy labeled samples, the inherent differences between visible and infrared modalities continue to hinder the model's feature extraction and matching capabilities. Consequently, in this section, we delve deeper into strategies to mitigate the discrepancies between these modalities, aiming to enhance the model's performance in SSVI-ReID tasks. To alleviate the impact of cross-modality on model performance, we can reduce the distance between each visible-infrared image pair with the same identity. Specifically, $N$ identities are randomly sampled from the dataset, and $P$ visible images and $P$ infrared images are sampled for each identity to form a mini-batch with $2 \times N \times P$ images. Then, to enhance the similarity between visible and infrared features, we define the following loss function:

$$L_{MCL-} = \frac{1}{N} \frac{1}{P} \sum_{n=1}^{N} \sum_{p=1}^{P} \left\| F^v_{n,p} - F^r_{n,p} \right\|, \tag{10}$$

**Figure 3: Illustration of the proposed MCL effects. The proposed method effectively reduces the distance between visible and infrared feature centers of the same identity by aligning them, thereby alleviating the impact of modality discrepancy on model performance. Different colors represent different identities. Different shapes represent different modality features.**

where $F_{n,p}^v$ and $F_{n,p}^r$ represent the normalized feature of the $p$-th visible image and infrared image of $n$-th identity respectively in each mini-batch.

However, due to semi-supervised settings, there are incorrect infrared pseudo-labels. Paired narrowing of the distance between visible and infrared images will further reduce the distance between indistinguishable erroneous infrared images and visible images, affecting model performance. What's more, although this paired loss will reduce the modality gap of cross-modality images, it may lead the network to focus more on some details, such as posture and accessories, rather than identity features. Based on this, we calculate the centers of the visible and infrared features of the same identity,

$$C_n^v = \frac{1}{P} \sum_{p=1}^P F_p^v, \qquad C_n^r = \frac{1}{P} \sum_{p=1}^P F_p^r, \qquad (11)$$

where $C_n^v$ and $C_n^r$ represent the centers of the visible and infrared features of the $n$-th identity respectively. By narrowing the distance between their centers, the modality gap between visible and infrared modalities can be narrowed, while avoiding the negative impact of a small amount of incorrectly labeled features. Therefore, the proposed modality consistency loss can be written as follows:

$$L_{MCL} = \frac{1}{N} \sum_{n=1}^N \left\| \phi(C_n^v) - \phi(C_n^r) \right\|, \qquad (12)$$

where $\phi(\cdot)$ is a linear kernel, variables are mapped to vectors in Hilbert Space through kernel functions. We project features onto Hilbert Space to measure the distance between them.

As shown in Figure 3, it is obvious that the optimization of MCL would make two modality features similar by bridging the modality gap by reducing the distance between the visible-infrared feature centers of the same identity. The proposed modality consistency loss not only reduces the modality discrepancy between visible and infrared images, but also narrows the feature gap within the same modality, encouraging the compact distribution of features with the same identity within each modality.

## 3.5 Optimization

The original visible and infrared images are fed together into the two-stream ResNet50 [14] backbone network, along with the generated intermediate features, to help optimize the network. In the proposed MUCG, in addition to the proposed $L_{WIL}$ and $L_{MCL}$, we also combined the triplet loss $L_{TRI}$ [16] and the adversarial loss $L_D$ [37] to jointly optimize the network together.

$$L_{BASE} = L_{TRI} + L_D, \qquad (13)$$

$L_{TRI}$ is used in VI-ReID tasks, as it helps to minimize intra-class similarity and maximize inter-class similarity in metric learning. $L_D$ is an adversarial loss in domain adaptation, assisting the model in learning modality-invariant features. The total loss of the proposed MUCG is defined as:

$$L_{MUCG} = L_{BASE} + \lambda_{WIL} L_{WIL} + \lambda_{MCL} L_{MCL}, \qquad (14)$$

where $\lambda_{WIL}$ and $\lambda_{MCL}$ are two trade-off hyper-parameters. Overall, the proposed method provides a comprehensive solution for SSVI-ReID, utilizing multiple loss functions and modalities to enhance the performance of the model.

## 4 EXPERIMENTS

### 4.1 Datasets

The proposed method is evaluated on three challenging VI-ReID datasets, *i.e.*, **SYSU-MM01** [40], **RegDB** [30], and **LLCM** [55]. The SYSU-MM01 dataset consists of 491 pedestrians with 287,628 visible images and 15,792 infrared images, captured by four visible and two infrared cameras. In addition, there are two search modes: all-search and indoor-search. The RegDB dataset consists of 412 pedestrian images captured by binocular cameras, each containing 10 thermal infrared images and 10 visible images. RegDB includes two testing settings: thermal to visible (IR to VIR) and visible to thermal (VIR to IR). The LLCM dataset consists of 1,064 identities captured by nine cameras deployed in low-light environments. Similar to the RegDB dataset, both the VIS to IR mode and the IR to VIS mode are used to evaluate the performance of the VI-ReID models.

**Evaluation Metrics.** The standard Cumulative Matching Characteristics (CMC) and the mean Average Precision (mAP) are used as the performance evaluation metrics in our experiments. For SYSU-MM01 and LLCM, we strictly follow the existing methods to select the gallery set for ten experiments [46, 55] and calculate the average performance value. For RegDB, We report the average result by randomly splitting of training and testing set 10 times [45].

### 4.2 Implementation Details

The proposed method is implemented with PyTorch. The model is trained for 80 epochs in total. We use ResNet-50 [14] pre-trained on ImageNet [7] as the backbone to extract image features. Following [55, 56], for the SYSU-MM01 dataset, the input images are resized to $384 \times 192$. In each mini-batch, we randomly select 4 visible images and 4 infrared images from 6 identities for training. For the RegDB and LLCM datasets, the input images are resized to $288 \times 144$. In each mini-batch, we randomly select 4 visible images and 4 infrared images from 8 identities for training. In the training stage, the input images are randomly flipped and erased with 50% probability [61], while visible images are extra randomly

**Table 1: Comparisons with state-of-the-art methods in different label-efficient VI-ReID on SYSU-MM01 (single-shot) and RegDB, *i.e.*, fully-supervised VI-ReID (SVI-ReID), unsupervised domain adaptation ReID (UDA-ReID), and semi-supervised VI-ReID (SSVI-ReID). All methods are measured by CMC (%) and mAP (%).**

| Settings | | | SYSU-MM01 | | | | RegDB | | | |
| --- | --- | --- | --- | --- | --- | --- | --- | --- | --- | --- |
| | | | All Search | | Indoor Search | | VIS to IR | | IR to VIS | |
| Type | Method | Venue | R-1 | mAP | R-1 | mAP | R-1 | mAP | R-1 | mAP |
| SVI-ReID | DDAG [48] | ECCV'20 | 54.8 | 53.0 | 61.0 | 68.0 | 69.3 | 63.5 | 68.1 | 61.8 |
| | AGW [49] | TPAMI'21 | 47.5 | 47.7 | 54.2 | 63.0 | 70.0 | 66.4 | 70.5 | 65.9 |
| | NFS [3] | CVPR'21 | 56.9 | 55.5 | 62.8 | 69.8 | 80.5 | 72.1 | 78.0 | 69.8 |
| | MID [19] | AAAI'22 | 60.3 | 59.4 | 64.9 | 70.1 | 87.5 | 84.9 | 84.3 | 81.4 |
| | FMCNet [54] | CVPR'22 | 66.7 | 62.5 | 68.2 | 74.1 | 89.1 | 84.4 | 88.4 | 83.9 |
| | DCLNet [35] | MM'22 | 70.8 | 65.2 | 73.5 | 76.8 | 81.2 | 74.3 | 78.0 | 70.6 |
| | PMT [27] | AAAI'23 | 67.5 | 65.0 | 71.7 | 76.5 | 84.8 | 76.6 | 84.2 | 75.1 |
| | ProtoHPE [53] | MM'23 | 71.9 | 70.6 | 77.8 | 81.3 | 88.7 | 83.7 | 88.7 | 82.0 |
| | DEEN [55] | CVPR'23 | 74.7 | 71.8 | 80.3 | 83.3 | 91.1 | 85.1 | 89.5 | 83.4 |
| | CAL [41] | ICCV'23 | 74.7 | 71.7 | 79.7 | 83.7 | 94.5 | 88.7 | 93.6 | 87.6 |
| | SAAI [9] | ICCV'23 | 75.9 | 77.0 | 83.2 | 88.0 | 91.1 | 91.5 | 92.1 | 92.0 |
| | PartMix [23] | CVPR'23 | 77.8 | 74.6 | 81.5 | 84.4 | 85.7 | 82.3 | 84.9 | 82.5 |
| UDA-ReID | MEB-Net [51] | ECCV'20 | 7.3 | 6.9 | 20.4 | 11.7 | 5.6 | 6.9 | 14.9 | 14.0 |
| | D-MMD [29] | ECCV'20 | 12.5 | 10.4 | 19.0 | 15.4 | 2.2 | 3.7 | 2.0 | 3.6 |
| | MMT [12] | ICLR'20 | 13.9 | 8.4 | 21.0 | 15.3 | 5.3 | 7.1 | 11.0 | 12.1 |
| | SpCL (UDA) [13] | NIPS'20 | 15.1 | 6.5 | 19.5 | 12.1 | 3.3 | 4.3 | 8.4 | 9.5 |
| | GLT [59] | CVPR'21 | 7.7 | 9.5 | 12.1 | 18.0 | 2.9 | 4.5 | 6.3 | 7.6 |
| | OTLA (UDA) [37] | ECCV'22 | 29.9 | 27.1 | 29.8 | 38.8 | 32.9 | 29.7 | 32.1 | 28.6 |
| | TAA with ResNet-50 [43] | TIP'23 | 40.6 | 33.3 | 41.5 | 47.1 | 58.5 | 53.2 | 57.5 | 52.0 |
| | TAA with AGW [43] | TIP'23 | 48.8 | 42.4 | 50.1 | 56.0 | 62.2 | 56.0 | 63.8 | 56.5 |
| SSVI-ReID | MAUM-50 [26] | CVPR'22 | 28.8 | 36.1 | - | - | - | - | - | - |
| | MAUM-100 [26] | CVPR'22 | 38.5 | 39.2 | - | - | - | - | - | - |
| | OTLA [37] | ECCV'22 | 48.2 | 43.9 | 47.4 | 56.8 | 49.9 | 41.8 | 49.6 | 42.8 |
| | DIPS [33] | ICCV'23 | 58.4 | 55.6 | 63.0 | 70.0 | 62.3 | 53.2 | 61.5 | 52.7 |
| | MUCG (ours) | - | **68.8** | **65.9** | **77.4** | **81.0** | **86.9** | **76.7** | **83.7** | **74.1** |

grayscale with 50% probability. The model is optimized by the Adam optimizer with an initial learning rate of $3.5 \times 10^{-3}$. The learning rate is incorporated with a warm-up strategy [28] and decayed 10 times at epoch 20 and epoch 50 [37]. The hyper-parameter $\lambda_{WIL}$ is set to 0.1. The hyper-parameter $\lambda_{MCL}$ is set to 5 on the SYSU-MM01 and LLCM datasets, and to 100 on the RegDB dataset.

## 4.3 Comparison with State-of-the-Art Methods under Various Settings

We compare our method with three related VI-ReID settings to demonstrate its effectiveness, *i.e.*, fully-supervised VI-ReID (SVI-ReID), unsupervised domain adaptation ReID (UDA-ReID), and semi-supervised VI-ReID (SSVI-ReID). Following [37], for UDA-ReID methods [12, 13, 29, 59], we use ground-truth labeled visible data as the source domain and unlabeled infrared data as the target domain. Following [43], for visible-infrared UDA-ReID methods [37, 43], we use other labeled visible data as the source domain and unlabeled VI-ReID data as the target domain. The experimental results on the SYSU-MM01 and RegDB datasets are reported in Table 1 and the results on the LLCM dataset are reported in Table 2. **Comparison with Fully-supervised Methods:** The proposed MUCG only with ground-truth visible data outperforms several fully supervised VI-ReID methods on the SYSU-MM01 and RegDB

**Table 2: Comparisons with state-of-the-art methods in different label-efficient VI-ReID on the LLCM dataset, *i.e.*, fully-supervised VI-ReID (SVI-ReID) and semi-supervised VI-ReID (SSVI-ReID). All methods are measured by CMC (%) and mAP (%). Method marked by † denotes re-implementations based on public code.**

| Settings | | | VIR to IR | | IR to VIS | |
| --- | --- | --- | --- | --- | --- | --- |
| Type | Method | Venue | R-1 | mAP | R-1 | mAP |
| SVI-ReID | DDAG [48] | ECCV'20 | 48.0 | 52.3 | 40.3 | 48.4 |
| | AGW [49] | TPAMI'21 | 51.5 | 55.3 | 43.6 | 51.8 |
| | LbA [31] | ICCV'21 | 50.8 | 55.6 | 43.8 | 53.8 |
| | CAJ [47] | ICCV'21 | 56.5 | 59.8 | 48.8 | 56.6 |
| | MMN [56] | MM'21 | 59.9 | 62.7 | 52.5 | 58.9 |
| | DART [44] | CVPR'22 | 60.4 | 63.2 | 52.2 | 59.8 |
| | DEEN [55] | CVPR'23 | 62.5 | 65.8 | 54.9 | 62.9 |
| SSVI-ReID | OTLA† [37] | ECCV'22 | 44.2 | 48.2 | 36.2 | 42.2 |
| | MUCG (ours) | - | **51.9** | **55.2** | **43.8** | **49.8** |

datasets and achieves comparative results on the LLCM dataset. The results indicate that the proposed MUCG can effectively utilize unlabeled infrared image information to improve model performance. However, there remains a certain gap between the proposed MUCG and the state-of-the-art fully supervised results.

**Table 3: Influence of each component on the performance of the proposed MUCG.**

| | Method | | | SYSU-MM01 | | RegDB | |
|---|---|---|---|---|---|---|---|
| Order | DIMG | WIL | MCL | R-1 | mAP | R-1 | mAP |
| 1 | | | | 43.6 | 42.5 | 66.2 | 59.6 |
| 2 | ✓ | | | 48.6 | 47.0 | 79.1 | 68.7 |
| 3 | ✓ | | ✓ | 53.8 | 50.8 | 80.8 | 70.6 |
| 4 | | ✓ | | 64.4 | 61.5 | 73.9 | 67.8 |
| 5 | | ✓ | ✓ | 64.9 | 62.4 | 84.0 | 75.5 |
| 6 | ✓ | ✓ | ✓ | **68.8** | **65.9** | **86.9** | **76.7** |

**Table 4: Influences of different weighted identification loss and different modality consistency loss.**

| Method | SYSU-MM01 | | RegDB | |
|---|---|---|---|---|
| | R-1 | mAP | R-1 | mAP |
| $L_{WIL-}$ | 67.5 | 64.0 | 86.6 | 75.4 |
| $L_{WIL}$ | **68.8** | **65.9** | **86.9** | **76.7** |
| $L_{MCL-}$ | 67.0 | 64.0 | 74.0 | 66.8 |
| $L_{MCL}$ | **68.8** | **65.9** | **86.9** | **76.7** |

**Comparison with Unsupervised Domain Adaptation Methods:** As we can see, the state-of-the-art UDA-ReID methods [12, 13] cannot achieve good results under semi-supervised VI-ReID settings due to the huge modality discrepancy. Although some UDA-ReID methods use stronger monitoring signals than ours, the accuracy is far lower than our method. On the other hand, UDA-VI-ReID [37] and [43] achieve better results than the traditional UDA-ReID [12] and [13]. This is because the traditional UDA-ReID method heavily relies on the labeled source domain, making the model less distinguishable for infrared data. Our MUCG can help the model alleviate modality gaps and achieve excellent performance. Specifically, compared to TAA [43], mAP gains of 23.5% and 20.7% are achieved on the SYSU-MM01 and RegDB datasets, respectively.

**Comparison with Semi-supervised Methods:** In the same experimental setting (SSVI-ReID), our method outperforms existing methods [48, 49]. Both OTLA [37] and DIPS [33] focus on handling infrared pseudo-labels, while neglecting the handling of the modality gap, and their handling of pseudo-labels is not comprehensive enough. OTLA focuses on generating pseudo-labels while neglecting the calibration of noisy labels. DIPS focuses on the calibration of noisy pseudo-labels. Compared with OTLA, our MUCG achieved 20.6% and 22.0% gains on the SYSU-MM01 dataset, 37.0% and 34.9% gains on the RegDB dataset, and 7.7% and 7.0% gains on the LLCM dataset, respectively in Rank-1 and mAP. MAUM-50 and MAUM-100 use 50 and 100 IR identities respectively to train the VI-ReID model. Our MUCG does not require IR data annotation and performs better than MAUM.

## 4.4 Ablation Studies

**The Influence of Different Components:** To evaluate the contribution of each component to MUCG, we conduct some ablation studies on the SYSU-MM01 dataset. The overall settings remain the same, while only the module under demonstration is added or removed from MUCG. As shown in Table 3, by incorporating

**Table 5: Effectiveness on which stage of ResNet-50 to plug DIMG into.**

| Method | SYSU-MM01 | | | |
|---|---|---|---|---|
| | R-1 | R-10 | R-20 | mAP |
| DIMG after stage-1 | 66.0 | 93.0 | 96.9 | 62.9 |
| DIMG after stage-2 | 66.2 | 93.2 | 97.1 | 63.1 |
| DIMG after stage-3 | **68.8** | **94.7** | **97.8** | **65.9** |
| DIMG after stage-4 | 68.0 | 94.4 | 97.7 | 64.4 |

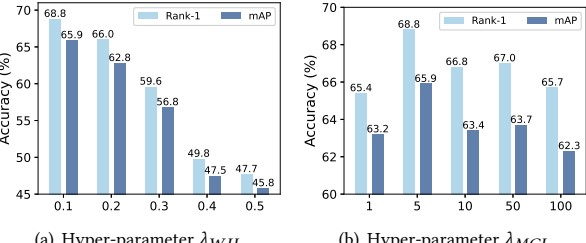

(a) Hyper-parameter $\lambda_{WIL}$      (b) Hyper-parameter $\lambda_{MCL}$

**Figure 4: Influence of different $\lambda_{WIL}$ and $\lambda_{MCL}$ on the SYSU-MM01 dataset.**

the proposed DIMG module into the backbone network, we can effectively enhance the ability to extract discriminative features and alleviate the visible-infrared modality discrepancy(see $1^{st}$ row and $2^{nd}$ row, $5^{th}$ row and $6^{th}$ row). The weighted processing of pseudo-labels by the WIL module greatly alleviates the negative impact of incorrect pseudo-labels on the model (see $1^{st}$ row and $4^{th}$ row). The MCL module can further reduce the modality discrepancy between visible and infrared features, ultimately improving the performance of the SSVI-ReID task (see $2^{nd}$ row and $3^{rd}$ row, $4^{th}$ row and $5^{th}$ row). Compared with the baseline, the proposed MUCG achieves gains of 25.2% and 23.4% in Rank-1 and mAP on the SYSU-MM01 dataset, respectively.

**The Influence of Different Weighted Identification Loss and Modality Consistency Loss:** To demonstrate that using low-confidence samples as negative samples can improve the WIL module, we conduct experiments to compare the results of using $L_{WIL-}$ and $L_{WIL}$. As shown in Table 4, it can be observed that when optimizing by $L_{WIL}$, the network achieves the best performance, surpassing the $L_{WIL-}$ by 1.3% and 1.9% on the SYSU-MM01 dataset, respectively in Rank-1 and mAP. To demonstrate that using feature centers of the same identity to measure the distribution of visible and infrared modalities is more effective than using one-to-one corresponding visible-infrared features to measure the distribution, we conduct experiments to compare the results of using $L_{MCL-}$ and $L_{MCL}$. As shown in Table 4, it can be observed that the network achieves the best performance when optimizing by $L_{MCL}$, surpassing the $L_{MCL-}$ by 1.8% and 1.9% on the SYSU-MM01 dataset, respectively in Rank-1 and mAP.

## 4.5 Further Analysis

**The Influence of Plugging DIMG Module at Different Stages of ResNet-50:** The proposed DIMG can be integrated into any stage

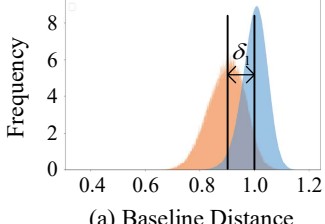
(a) Baseline Distance

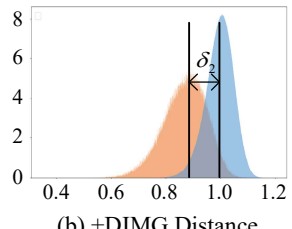
(b) +DIMG Distance

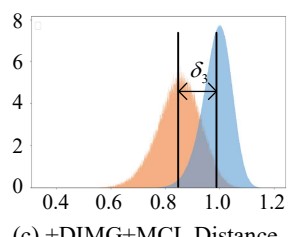
(c) +DIMG+MCL Distance

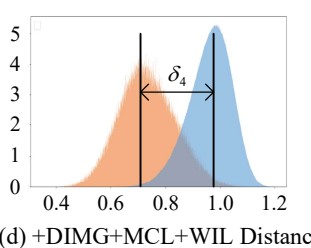
(d) +DIMG+MCL+WIL Distance

**Figure 5: The frequency of intra-class and inter-class distances between the cross-modality features of SYSU-MM01. The intra-class and inter-class distances are indicated in orange and blue colors, respectively.**

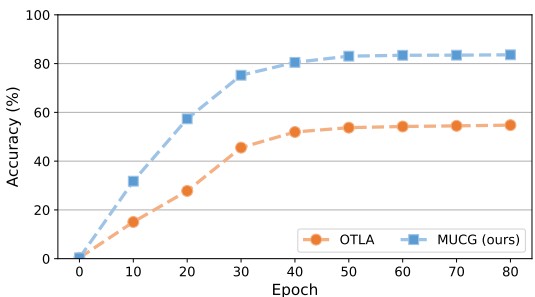

**Figure 6: Comparison of pseudo-label accuracy between the proposed MUCG and OTLA on the SYSU-MM01 dataset under the semi-supervised setting.**

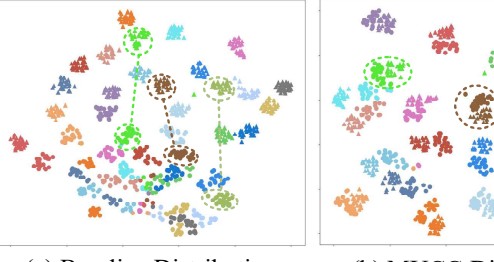
(a) Baseline Distribution          (b) MUCG Distribution

**Figure 7: The distribution of feature embeddings in the 2D feature. A total of 20 persons are selected from the test set. The samples with the same color are from the same person. The circle represents the visible modality and the triangle represents the infrared modality.**

of the backbone network. In our experiments, we use ResNet-50 as the backbone. We plug DIMG after different stages of the ResNet-50 to investigate how it affects the overall performance. As shown in Table 5, DIMG after stage-3 can achieve the best performance, which indicates that after stage-3, the proposed DIMG can better transfer visible knowledge to the infrared domain.

**The Influence of the Hyper-parameters** $\lambda_{WIL}$ **and** $\lambda_{MCL}$**:** To evaluate the influence of the two hyper-parameters, we give quantitative comparisons and report the results in Figure 4. As we can see, the best performance is achieved when $\lambda_{WIL}$ is set to 0.1 and $\lambda_{MCL}$ is set to 5, respectively.

**Pseudo-label Analysis:** We conduct an analysis experiment to evaluate the accuracy of pseudo-labels. As shown in Figure 6, as the training continues, the pseudo-label accuracy of the semi-supervised setting is iteratively improved. It can achieve an accuracy of 83.6% on the SYSU-MM01 dataset, surpassing OTLA's [37] 54.8%. Compared with OTLA, we penalize noisy labels while improving the model's discrimination ability for infrared images. As pseudo-labels are generated through model prediction, enhancing the performance of the model will significantly boost the accuracy of these labels.

### 4.6 Visualization

To investigate the reasons for the effectiveness of MUCG, we visualize inter-class and intra-class distances on the SYSU-MM01 dataset, as shown in Figure 5. Comparing Figure 5 (b-d) with (a), the means of inter-class and intra-class distances (*i.e.*, vertical lines) are pushed away by DIMG, MCL, and WIL, where $\delta_1 < \delta_2 < \delta_3 < \delta_4$. Figure

5 shows that the intra-class distances of MUCG are significantly smaller compared to the distances of baseline features. Therefore, MUCG can effectively reduce the distances between visible and infrared images. To further validate the effectiveness of the proposed MUCG, we plot the t-SNE distribution of the MUCG feature representations in the 2D feature space for visualization. As shown in Figure 7 (a) and 7 (b), the proposed MUCG method can significantly shorten the distance between images corresponding to the same identity in visible and infrared modalities, and effectively reduce modality discrepancy.

### 5 CONCLUSION

In this paper, we investigate the semi-supervised visible-infrared re-identification (SSVI-ReID) task, which can reduce the cost of cross-modality annotation. We propose a novel modality-unified and confidence-guided semi-supervised VI-Reid learning framework. We have also proposed three modules: DIMG, WIL, and MCL. DIMG can dynamically generate appropriate intermediate modality features, which helps improve the model's discrimination ability in the infrared domain and reduce modality discrepancies between visible and infrared modalities. In addition, we use the WIL to reduce the negative impact of incorrect labels on the model, and we use the MCL to narrow the distance between visible and infrared modality features. Extensive experiments have shown that MUCG outperforms the state-of-the-art semi-supervised methods and some fully supervised methods.

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
