# OpenReview forum: "Semi-supervised Visible-Infrared Person Re-identification via Modality Unification and Confidence Guidance"
_acmmm.org/ACMMM/2024/Conference — MM2024 Poster_

### Official Review · Reviewer_FhnM · 2024-05-01

**Rating:** 3
**Confidence:** 4

**Summary:**

This paper focuses on semi-supervised VI-ReID tasks. A DIMG module is proposed to bridge the modality gap at the feature level by dynamically generating intermediate features. A WIL loss is proposed to assign different weights to instances based on the confidence of their pseudo-labels. Additionally, an MCL loss is introduced to narrow the modality gap between cross-modality centroids.

**Strengths:**

(1) The proposed method achieves remarkable performance, which surpasses the SOTAs by a considerable margin;\
(2) The extensive experiments demonstrate the effectiveness of each component of the proposed method. The experiments involving the detailed design of each module are comprehensive. The illustrations and visualizations are appropriate.

**Limitations:**

(1) The intermediate modality generation strategy is commonly employed in existing methods [1,2]. It is important to clearly elucidate the distinctions and advantages of DIMG compared to these methods.\
(2) The confidence-guided WIL loss appears to be inspired by [3]. The main novelty lies in the improvement from WIL to WIL. Nevertheless, the extent of this improvement seems limited.\
(3) This paper integrates PCB [4] into their baseline. It might be unfair to compare with other methods that do not utilize PCB. Therefore, the authors are suggested to provide the performance gain achieved by PCB.

[1] Yu H, Cheng X, Peng W, et al. Modality unifying network for visible-infrared person re-identification[C]//Proceedings of the IEEE/CVF International Conference on Computer Vision. 2023: 11185-11195.\
[2] Zhang Y, Yan Y, Lu Y, et al. Towards a unified middle modality learning for visible-infrared person re-identification[C]//Proceedings of the 29th ACM international conference on multimedia. 2021: 788-796.\
[3] Shi J, Zhang Y, Yin X, et al. Dual pseudo-labels interactive self-training for semi-supervised visible-infrared person re-identification[C]//Proceedings of the IEEE/CVF International Conference on Computer Vision. 2023: 11218-11228.\
[4] Sun Y, Zheng L, Yang Y, et al. Beyond part models: Person retrieval with refined part pooling (and a strong convolutional baseline)[C]//Proceedings of the European conference on computer vision (ECCV). 2018: 480-496.\

**Suitability:**

3

---

### Official Review · Reviewer_UE8J · 2024-05-13

**Rating:** 5
**Confidence:** 4

**Summary:**

This paper proposes a Modality Unification and Confidence Guidance (MUCG) semi-supervised VI-ReID method, addressing pseudo-label errors and modality differences in existing work. The proposed Dynamic Intermediate Modality Generation (DIMG) module effectively transfers annotated visible image knowledge to unlabeled infrared images. Meanwhile, by introducing Weighted Identification Loss (WIL) and Modality Consistency Loss (MCL), the robustness and performance of the model are further improved. The experimental part also fully verified the effectiveness of the proposed method, which has significant advantages compared to existing methods.

**Strengths:**

1. From the content of the paper, it appears that the logic is clear and the structure is rigorous. The DIMG module can effectively solve the problems of pseudo label errors and modal differences. The proposal of WIL and MCL has further improved the performance of the model.
2. In the experimental section, the authors conduct sufficient validation and compare it with existing methods. The experimental results show that the proposed method has achieved significant performance improvement on semi-supervised VI-ReID tasks, fully demonstrating its effectiveness and superiority.

**Limitations:**

1. When describing the DIMG module, its working principle and implementation details can be further elaborated. For example, explain how it enhances the quality of pseudo labels through knowledge transfer, and how it bridges modal differences.
2. Please explain why MCL utilizes feature centers to reduce intra-class variance, instead of operating directly on individual sample features. What advantages or considerations underlie this design choice?
3. In the title of Figure 7, please indicate the specific dataset used for the experiment depicted in the figure for clarity.
4. There is a small writing error: in Table. 2, the "VIR" should be written as "VIS".

**Suitability:**

3

---

### Official Review · Reviewer_WNYe · 2024-05-19

**Rating:** 5
**Confidence:** 4

**Summary:**

This paper introduces a Modality-Unified and Confidence-Guided (MUCG) semi-supervised VI-ReID framework. The authors propose three modules - Dynamic Intermediate Modality Generation (DIMG), Weighted Identification Loss (WIL), and Modality Consistency Loss (MCL), addressing the issues of erroneous pseudo-labels and large modality discrepancy between visible and infrared images. Compared with existing methods, the performance is significantly improved.

**Strengths:**

1. This paper introduces a DIMG module that effectively bridges the gap between visible and infrared modalities, and enhances the model's discriminative ability for infrared modality, thereby elevating the pseudo-label quality and narrowing the modality gap.
2. This paper also introduces WIL and MCL. WIL can reduce the model's dependency on erroneous labels through confidence weighting, while MCL further minimizes the distribution gap between visible and infrared features, enabling the model to learn modality-unified features.
3. Experimental results demonstrate that the proposed MUCG method achieves significant performance improvements on the semi-supervised VI-ReID task, surpassing current state-of-the-art methods. This indicates that the proposed method is highly effective and practical in addressing the semi-supervised VI-ReID problem.

**Limitations:**

1. The Fig.1 lacks explanations, which may confuse readers when understanding the content of the figure. Suggest the authors add annotations in Figure 1 and clearly explain the meaning and direction of the arrows so that readers can have a more intuitive understanding of the process and structure of the model.
2. What is the calculation method for confidence weight in WIL? How to determine appropriate weight values to ensure that the model can balance the contribution of samples with different confidence levels during the training process?
3. Although MCL helps to narrow the modality gap, it remains unclear whether it can maintain sufficient robustness in the presence of a large number of erroneous labels. Is it necessary to combine other techniques (such as data cleaning, label correction, etc.) to further enhance the model's performance?
4. In the experimental section, apart from demonstrating performance improvements, it would be beneficial to include an analysis of the characteristics of different datasets and discuss the key factors that influence performance, thereby enhancing the persuasiveness of the experimental results.

**Suitability:**

3

---

### Official Review · Reviewer_crwz · 2024-05-23

**Rating:** 4
**Confidence:** 3

**Summary:**

The article presents a semi-supervised visible-infrared person re-identification (SSVI-ReID) approach called the Modality-Unified and Confidence-Guided (MUCG) framework. This framework is designed to address the challenges of erroneous pseudo-labels and significant modality discrepancies by introducing innovative components. The Dynamic Intermediate Modality Generation (DIMG) module creates intermediate modality features by blending features from visible and infrared images to bridge the gap between modalities and enhance the discriminative capability for unlabeled infrared images. The Weighted Identification Loss (WIL) leverages the memory effect of deep neural networks to calculate the correct labeling confidence for each sample, reducing the negative impact of incorrect labels. The Modality Consistency Loss (MCL) further minimizes the feature distribution gap between visible and infrared modalities by calculating the distance between feature centers of the same identity, optimizing feature similarity across modalities. The experiments conducted on multiple datasets demonstrate that MUCG surpasses current state-of-the-art methods in SSVI-ReID tasks, offering an effective semi-supervised learning strategy, especially beneficial for scenarios with limited annotation resources. This work advances research in cross-modality person re-identification and provides new insights and tools for future studies in related fields.

**Strengths:**

1. This paper innovatively combines modality unification and confidence-guided learning to address the unique challenges of semi-supervised visible-infrared person re-identification. The introduction of the Dynamic Intermediate Modality Generation (DIMG) module is particularly novel, as it effectively generates intermediate features that bridge the gap between visible and infrared modalities, enhancing the model's capability to handle unlabelled infrared images.
2. The theoretical approach is solid and well-founded, integrating the DIMG with Weighted Identification Loss (WIL) and Modality Consistency Loss (MCL) to systematically reduce the impact of erroneous pseudo-labels and modality discrepancies. This combination ensures the model can learn more robust and unified features, which is crucial for cross-modality tasks.
3. The paper demonstrates correctness and thoroughness through comprehensive experiments. The evaluation is extensive, with the MUCG framework being tested on multiple challenging datasets (SYSU-MM01, RegDB, LLCM), and consistently outperforming existing state-of-the-art methods. The use of standard evaluation metrics (Cumulative Matching Characteristics and mean Average Precision) further validates the robustness and applicability of the proposed approach.

**Limitations:**

Although the Weighted Identification Loss (WIL) aims to mitigate the impact of noisy labels, the reliance on pseudo-labels still introduces a vulnerability, especially if the initial pseudo-labels are of low quality. This can lead to a propagation of errors, which is a common challenge in semi-supervised learning frameworks.
2. the paper could improve in terms of clarity and depth of certain methodological explanations. For example, the specifics of how the Dynamic Intermediate Modality Generation (DIMG) module operates could be elaborated more comprehensively to provide better insights into its inner workings and effectiveness. This would help in understanding the potential limitations or constraints of this module in different contexts.
3. The paper does not delve deeply into the computational complexity or efficiency of the proposed method. Given that practical applications often require real-time processing, an analysis of the computational overhead introduced by the DIMG, WIL, and MCL modules would have been valuable. Without this, it is difficult to gauge the method's feasibility for deployment in resource-constrained environments.

**Suitability:**

3

---

### Meta-Review · Area_Chair_gfQr · 2024-06-27

**Recommendation:** Accept (Poster)
**Confidence:** 5

**Metareview:**

This paper proposes a Modality Unification and Confidence Guidance (MUCG) method for semi-supervised visible-infrared person re-identification (SSVI-ReID), addressing pseudo-label errors and modality differences. The Dynamic Intermediate Modality Generation (DIMG) module creates intermediate modality features by blending features from visible and infrared images to bridge the gap between modalities and enhance the discriminative capability for unlabeled infrared images. The Weighted Identification Loss (WIL) leverages the memory effect of deep neural networks to calculate the correct labeling confidence for each sample, reducing the negative impact of incorrect labels. The Modality Consistency Loss (MCL) further minimizes the feature distribution gap between visible and infrared modalities by calculating the distance between feature centers of the same identity, optimizing feature similarity across modalities. Extensive experimental results show that the proposed method has achieved significant performance improvement on semi-supervised VI-ReID tasks.

Initially, this paper received three positive scores and one negative review. After the rebuttal, all the reviewers have updated the score to Weak Accept. Therefore, I would like to accept this paper. The authors may carefully prepare the final camera ready version according to the reviewer comments. Some writing issues could be improved and missing references should be added.

---

### Meta-Review · Senior_Area_Chairs · 2024-07-10

**Recommendation:** Accept (Poster)
**Confidence:** 4

**Metareview:**

This paper received mixed ratings initially. After rebuttal, all the reviewers tend to accept the paper. SAC and AC agree with reviewers and recommend acceptance of the paper.